# Quantum chaos in $\mathcal{PT}$-symmetric quantum systems

Kshitij Sharma[1*], Himanshu Sahu [1,2†] and Subroto Mukerjee[1‡]

**1** Department of Physics, Indian Institute of Science, C.V. Raman Avenue, Bangalore 560012, India.

**2** Department of Instrumentation & Applied Physics, Indian Institute of Science, C.V. Raman Avenue, Bangalore 560012, Karnataka, India.

\* kshitijvijay@iisc.ac.in , † himanshusah1@iisc.ac.in , ‡ smukerjee@iisc.ac.in

July 25, 2025

## Abstract

In this study, we explore the interplay between $\mathcal{PT}$-symmetry and quantum chaos in a non-Hermitian dynamical system. We consider an extension of the standard diagnostics of quantum chaos, namely the complex level spacing ratio and out-of-time-ordered correlators (OTOCs), to study the $\mathcal{PT}$-symmetric quantum kicked rotor model. The kicked rotor has long been regarded as a paradigmatic dynamic system to study classical and quantum chaos. By introducing non-Hermiticity in the quantum kicked rotor, we uncover new phases and transitions that are absent in the Hermitian system. From the study of the complex level spacing ratio, we locate three regimes – one which is integrable and $\mathcal{PT}$-symmetry, another which is chaotic with $\mathcal{PT}$-symmetry and a third which is chaotic but with broken $\mathcal{PT}$-symmetry. We find that the complex level spacing ratio can distinguish between all three phases. Since calculations of the OTOC can be related to those of the classical Lyapunov exponent in the semi-classical limit, we investigate its nature in these regimes and at the phase boundaries. In the phases with $\mathcal{PT}$-symmetry, the OTOC exhibits behaviour akin to what is observed in the Hermitian system in both the integrable and chaotic regimes. Moreover, in the $\mathcal{PT}$-symmetry broken phase, the OTOC demonstrates additional exponential growth stemming from the complex nature of the eigenvalue spectrum at later times. We derive the analytical form of the late-time behaviour of the OTOC. By defining a normalized OTOC to mitigate the effects caused by $\mathcal{PT}$-symmetry breaking, we show that the OTOC exhibits singular behaviour at the transition from the $\mathcal{PT}$-symmetric chaotic phase to the $\mathcal{PT}$-symmetry broken, chaotic phase.

# 1   Introduction

Over the years, the notion of chaos in a quantum system has been placed on a firm footing despite the absence of a phase space in which to describe the dynamics, [1, 2]. Diagnostics such as the energy level spacing distribution have been employed to characterize chaotic quantum Hamiltonians due to their similarities with random matrices [3, 4]. Random Matrix Theory (RMT) is a powerful tool that accurately describes the spectral statistics of quantum systems whose classical counterparts exhibit chaotic behaviour. In cases where quantum Hamiltonians correspond to integrable classical systems, the Berry-Tabor conjecture proposes that their level spacing distributions have Poissonian forms, [5]. On the other hand, for quantum Hamiltonians with chaotic classical counterparts, the Bohigas-Giannoni-Schmit conjecture proposes that the level statistics should correspond to one of the three classical ensembles of RMT (Wigner-Dyson ensembles) [6], namely – the Gaussian unitary ensemble (GUE), the Gaussian orthogonal ensemble (GOE), and the Gaussian symplectic ensemble (GSE) that come out of Wigner's surmise, [7, 8]. Wigner's surmise is a statement about the probability density of nearest-neighbor energy level spacings of Hermitian random matrices. Based on the symmetries of the matrix, the probability density takes one of three forms which can then be used to deduce information about the symmetries from the energy level spacing distribution. A similar diagnostic is the energy level spacing ratio, which has proven more versatile due to its independence from local energy densities [9].

The classical kicked rotor has proven paradigmatic in understanding chaos classically in time-dependent 1D systems. It can be stroboscopically evolved using the Chirikov standard map [10] and exhibits a transition from integrability to chaos with increasing kicking strength [11]. The quantum version of the classical kicked rotor also displays a transition from being integrable to chaotic [12–14]. This paper explores chaos in a non-Hermitian version of the quantum kicked rotor.

In recent studies, it has been observed that the out-of-time-order correlator (OTOC) carries information on chaos in quantum systems [15]. The Lyaponuv exponent, which is a classical measure of chaos, can be extracted from taking derivatives of the canonical variables with respect to initial conditions. When such a derivative is expressed as a Poisson bracket, and the brackets are transformed into commutators, one obtains the OTOC [16].

The out-of-time-order correlator (OTOC) [16–19] is another possible measure of quantum chaos that may be used to identify an analogue of the Lyapunov exponent, providing a connection with classical chaos, e.g., via the butterfly effect. Previously, OTOCs have been calculated in the context of information scrambling [20–23], quantum butterfly effects [24], many-body localization [25], "fast scrambling" [26–28], dynamical and topological phase transitions [29–32], and open quantum systems [33, 34]. Recently, the experimental implementation of many-body time-reversal protocols in atomic quantum systems has attracted attention for its potential to measure OTOCs experimentally [35–38] leading to several specific experimental proposals to measure OTOCs and also the first experimental demonstrations [15, 39–41]

The OTOC has been calculated for the Hermitian kicked rotor [42]. It has been observed that at initial times, the OTOC exhibits exponential growth in the chaotic regime, which is absent in the integrable regime. Therefore, it has been possible to extract a quantum equivalent of the classical Lyapunov exponent, at least in a semi-classical limit.

Non-Hermitian Hamiltonians have been studied extensively over the last few years due to the discovery of phenomena such as the non-Hermitian skin effect [43], the presence of exceptional points [44] and interesting topological properties in these systems [45, 46]. The study of the effect of electron-electron interactions in non-Hermitian systems with Lorentz symmetry [47–50] also proves to be of interest as it leaves the eigenspectrum of the system completely real Bender showed that if a Hamiltonian commutes with the operator $\mathcal{PT}$ where $\mathcal{P}$ is the (unitary) parity operator and $\mathcal{T}$ is the (anti-unitary) time reversal operator, it can possess a completely real spectrum of eigenvalues without necessarily being Hermitian [51, 52]. In fact, one typically chooses a parameter to tune in such a $\mathcal{PT}$-symmetric Hamiltonian to go from a $\mathcal{PT}$-symmetric phase to a $\mathcal{PT}$-symmetry broken phase, in which the energy eigenvalues are complex [51].

It is thus interesting to investigate whether a system can exhibit both an integrable to chaotic transition and a $\mathcal{PT}$-symmetry-breaking transition, and if so, what are the possible resultant phases? Motivated by the above question, in this work, we study a non-Hermitian extension of the kicked rotor model, which we henceforth refer to as the $\mathcal{PT}$-symmetric kicked rotor (PTKR) model.

Since the PTKR model is non-Hermitian and may have complex eigenvalues, we must use generalizations of the previously discussed diagnostics of quantum chaos [53] that apply to complex eigenvalues. For example, we study the complex energy level spacing ratio (CLSR) [54, 55] instead of the conventional real energy level spacing ratio (RLSR) [9]. Additionally, we calculate a suitably defined OTOC for non-Hermitian systems. [56]. Recently, the time evolution of the OTOC has been studied for the PTKR model and its classical counterpart in the vicinity of a $\mathcal{PT}$ symmetry-breaking transition in parameter space. [56–58]. Our work has also been inspired by previous work on the Hermitian-kicked rotor [42], which presents a numerical calculation of the Out of Time Ordered Correlator (OTOC) to the study the transition from integrability to chaos in the model. Our study extends this study to include the entire phase diagram of the PTKR model. It provides critical insights into the interplay of the $\mathcal{PT}$-symmetry-breaking transition and the transition from integrability to chaos. We study the interplay of these transitions by calculating the Complex level Spacing Ratio (CLSR) introduced in [54]. Additionally, we study a modified version of the OTOC by taking into account the lack of unitarity in the non-Hermitian systems to show possible markers of these transitions that can be captured by the OTOC.

The rest of the paper is structured as follows. In section 2, we introduce the PTKR model. In section 3, we discuss the two diagnostics of quantum chaos – complex level spacing ratio and out-of-ordered correlator. The main results based on these diagnostics are presented in

133 section 4. Finally, we conclude in section 5 discussing our work's implications and future
134 directions.

## 2 Model

136 We modify the Hermitian quantum kicked rotor model by adding a non-Hermitian term that
137 preserves $\mathcal{PT}$-symmetry. The resultant model is the PTKR model described by the Hamiltonian

$$H = \frac{p^2}{2m} + V(\theta) \sum_n \delta \left( n - \frac{t}{\tau} \right) \tag{1}$$

138 where

$$V(\theta) = K \frac{(\cos\theta + i\lambda\sin\theta)}{\sqrt{1+\lambda^2}} . \tag{2}$$

139 Here $\theta$ is the angle the rotor makes with a pre-defined direction and the momentum $p = -i\hbar_{\text{eff}} d/$
140 $d\theta$. When set to zero, the non-Hermiticity parameter $\lambda$ gives the Hermitian model. The action
141 of the parity operator $\mathcal{P}$ is defined as follows-

$$\mathcal{P}\hat{\theta}\mathcal{P}^{-1} = -\hat{\theta} \qquad \mathcal{P}\hat{p}\mathcal{P}^{-1} = -\hat{p} \tag{3}$$

142 The time reversal operator $\mathcal{T}$ is just the complex conjugation operator. Thus, we have that
143 $[H, \mathcal{PT}] = 0$ but $[H, \mathcal{P}] \neq 0$ and $[H, \mathcal{T}] \neq 0$. The PTKR model Hamiltonian is time-dependent
144 and, therefore, cannot be diagonalized by a time-independent transformation. However, since
145 the time dependence is periodic with a constant period $\tau$, we can define a Floquet evolution
146 operator of the time-independent Floquet Hamiltonian $\mathcal{H}_{\mathcal{F}}$.

147 The Floquet evolution operator is usually a unitary operator defined over a fixed time
148 period of the system, providing a stroboscopic view of the system's evolution. For our Hamil-
149 tonian, the Floquet operator is no longer unitary. We define it as follows

$$\mathcal{F}_{t_0} = T \exp\left\{ \frac{-i}{\hbar} \int_{t_0}^{\tau + t_0} H(t) dt \right\} \tag{4}$$

150 where $T$ stands for time ordering. The form of the operator thus becomes

$$\mathcal{F} = \exp\left\{ -i\frac{p^2}{4} \right\} \exp\{-iV(\theta)\} \exp\left\{ -i\frac{p^2}{4} \right\} \tag{5}$$

151 where we have taken $\hbar_{\text{eff}} = 1, \tau = 1$ and $m = 1$. The above-defined operator is time-
152 independent and can be diagonalized to obtain its eigenvalues. To extract the eigenvalues
153 of the PTKR model Hamiltonian from its Floquet evolution operator, we take the logarithm of
154 the operator's eigenvalues and multiply them by $i$. Due to the periodic nature of phases, the
155 eigenvalue spectrum of the Floquet Hamiltonian is folded into a single interval modulo the
156 period $2\pi$. Since this procedure causes eigenvalues that would otherwise have been far apart
157 to now be proximate, it generates chaos.

## 3  Methods

### 3.1  Energy Level Spacing Ratio

A quantity derivable from the eigenvalue spectrum of a random matrix is the level spacing ratio ($r$). For the case of a real energy spectrum, it is defined as

$$r = \frac{1}{N_M - 2} \sum_{\beta=1}^{N_M - 2} \frac{\min(\mu_{\beta+2} - \mu_{\beta+1}, \mu_{\beta+1} - \mu_\beta)}{\max(\mu_{\beta+2} - \mu_{\beta+1}, \mu_{\beta+1} - \mu_\beta)}. \tag{6}$$

In the above equation, $\mu_i$ is the $i^{\text{th}}$ eigenvalue out of a total $N_M$ eigenvalues arranged in ascending order. The advantage $r$ has over calculating the level spacing distribution is that it does not require an unfolding procedure (i.e. a normalization of the level spacing distribution by the local density of states). Quantum chaotic systems possess the same energy level spacing distribution as random Hamiltonians with the same set of symmetries [3]. To extend the analysis of the energy level spacing distribution to non-Hermitian systems, we use the complex level spacing ratio $\xi$ for the $\gamma^{\text{th}}$ state [54] defined below where $z_\gamma$ is its complex eigenvalue.

$$\xi_\gamma = \frac{z_\gamma^{\text{NN}} - z_\gamma}{z_\gamma^{\text{NNN}} - z_\gamma} = r_\gamma \exp\{i\theta_\gamma\} \tag{7}$$

The complex level spacing ratio (CLSR) can be defined in two ways: One, by averaging over $\gamma$ the magnitudes of $\xi_\gamma$ to obtain $\langle r \rangle$ (derived from the set of $r_\gamma$ values), and the other, by averaging over the angular distribution to obtain $-\langle \cos\theta \rangle$ (derived from the set of $\theta_\gamma$ values). Only $\langle r \rangle$ can be defined in the limit when the spectrum becomes real. However, in this limit, Eq. 7 does not yield the standard real level spacing ratio (RLSR) for a real spectrum. We, henceforth, refer only to the mean value $\langle r \rangle$ as the complex level spacing ratio (CLSR). Calculations of $-\langle \cos\theta \rangle$ can be found in the supplementary material. The universality classes for random matrices that describe Hermitian matrices in terms of Gaussian ensembles have suitable generalization to complex matrices [53]. We calculate the CLSR for these universality classes, one of which the PTKR model belongs.

To improve the statistics in our calculation, we replace $m$ in Eq. (1) by $m + \Delta m_p$, where $\Delta m_p$ is a small random number selected independently for each momentum eigenstate $|p\rangle$. This ensures that no underlying symmetries leading to unwanted degeneracies occur. This is important as degeneracies must be avoided when calculating level spacing ratios.

### 3.2  Out-of-time-order correlator

Recently, the study of OTOCs has extended to non-Hermitian systems, particularly those that possess $\mathcal{PT}$-symmetry. In the context of the QKR, these studies have shown that OTOC increases as a power law in time in the broken $\mathcal{PT}$-symmetry phase [56, 57, 59]. In this study, we investigate the nature of the OTOC in the different phases identified employing the complex level spacing ratio. The OTOC is typically defined as

$$C(t) \equiv -\langle [W(t), V(0)]^2 \rangle, \tag{8}$$

where $\langle \cdots \rangle$ represents the expectation values with respect to a state $|\Psi\rangle$. $W(t)$ and $V(t)$ are operators at time $t$ in the Heisenberg representation. In what follows, we choose $W(t) = \hat{p}(t)$, and $V(0) = \hat{p}(0)$ in Eq. (8). The state $|\Psi\rangle$ is a Gaussian wave packet

$$|\Psi\rangle = \sum_{k=-\infty}^{\infty} a_k^{(0)} |k\rangle \tag{9}$$

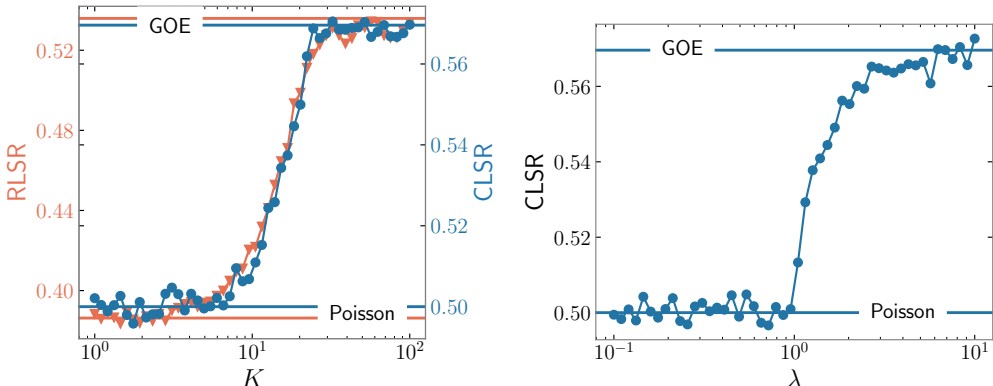

Figure 1: **Left** The CLSR and RLSR as functions of the kicking strength $K$, calculated for $N = 10095$ with $\hbar_{\text{eff}} = 0.2$ and $\lambda = 0$ (Hermitian). Horizontal lines in orange (blue) correspond to values of the RLSR (CLSR) for Poisson and GOE statistics. The figure shows that the CLSR is as good an indicator of the transition from integrability to chaos as the more commonly employed RLSR. **Right** The CLSR as a function of the non-Hermiticity parameter $\lambda$, calculated for $N = 10095$ with $\hbar_{\text{eff}} = 0.2$ and $K = 0.15$. Horizontal lines represent the values of CLSR for the Poisson and GOE distributions. The figure resembles the transition observed for the CLSR on the left obtained by varying $K$ with $\lambda = 0$. However, it is obtained by tuning only $\lambda$ instead of $K$. This suggests that the system can transition from a $\mathcal{PT}$-symmetric integrable phase to a $\mathcal{PT}$-symmetric chaotic phase with increasing non-Hermiticity while the value of $K$ is kept constant.

where

$$a_k^{(0)} \sim \exp\left(-\frac{\hbar_{\text{eff}}^2(k-k_0)^2}{2\sigma^2}\right), \qquad \hat{p}|k\rangle = \hbar_{\text{eff}}k|k\rangle .$$

We choose $p_0 = \hbar_{\text{eff}}k_0 \in [-\pi, \pi]$ and $\sigma = 4$. Numerically, $|\Psi\rangle$ is represented in a finite basis of eigenstates $|k\rangle$, $k \in [-N; N-1]$. The Hamiltonian is expressed as a matrix in this basis of $|k\rangle$ states, and all calculations are also performed w.r.t this basis. In both, the calculation of the OTOC and CLSR, an "effective value" of the Planck constant $\hbar_{\text{eff}}$ has been employed as a parameter to tune the amount of discreteness of the momentum operator [42]. This allows one to make a connection with the classical model by taking the limit $\hbar_{\text{eff}} \to 0$.

## 4 Results

We now present the results of the calculations discussed in Sec. 3 for the PTKR model.

### 4.1 CLSR

For the Hermitian case ($\lambda = 0$), the mean RLSR can be used to identify the transition from integrability to chaos. In what follows, we show that the CLSR can also be used for the same purpose and takes on characteristic values in the two regimes.

In Fig. 1, we show the mean RLSR as well as the CLSR with varying values of the kicking strength $K$, for $\hbar_{\text{eff}} = 0.2$ and system-size $N = 10095$. We notice that the transition points are reasonably independent of the value of $N$. Thus, in what follows, the system size is chosen to be large enough so that all quantities are well converged. We find that both the CLSR and RLSR display a transition at the same value of $K$. However, the values in the two regimes

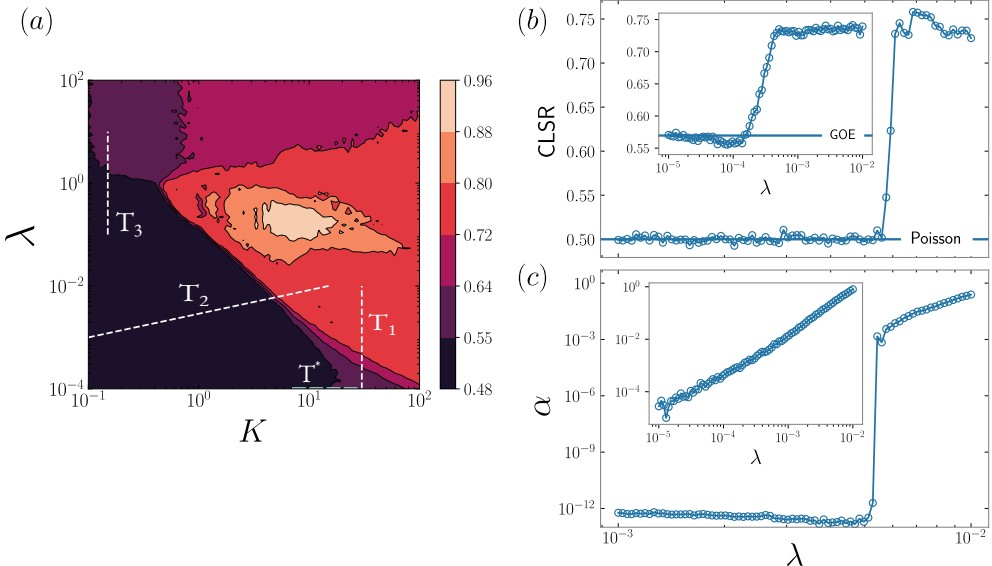

Figure 2: **(a)** The CLSR for varying values of the kicking strength $K$ and the non-Hermiticity $\lambda$, computed for $N = 6005$, $\hbar_{\text{eff}} = 0.2$. The dashed lines $T_2$ and $T_1$ represent the transitions across the $\mathcal{PT}$-symmetric integrable phase to $\mathcal{PT}$-symmetry broken chaotic phase and the $\mathcal{PT}$-symmetric chaotic phase to $\mathcal{PT}$-symmetry broken chaotic phase, respectively. Line $T^*$ is the Hermitian transition from the integrable to chaotic phase. $T_3$ marks a transition similar to $T^*$ in which $K$ is held constant and $\lambda$ varied. **(b)**: The CLSR across the transition **Main panel**: $T_2$ **Inset** $T_1$, calculated for $N = 6005$. The horizontal lines show the standard values of the CLSR for the GOE and Poisson distributions. **(c)**: The maximum imaginary part of an energy eigenvalue, $\alpha$, across the transition **Main panel**: $T_2$ **Inset** $T_1$, calculated for $N = 6005$. It can be seen that while $\alpha$ shows an abrupt change along $T_2$, it seems to increase smoothly along $T_1$. The CLSR on the other hand, shows an abrupt transition along both $T_2$ and $T_1$.

Table 1: The table below consists of data gathered on CLSR and RLSR of standard known values in the integrable regime and random matrix ensemble calculations in the chaotic regime [9, 55].

| Universality Class | RLSR | CLSR |
|---|---|---|
| Poisson | 0.386 | 0.50 |
| GOE | 0.536 | 0.57 |

are different. We can understand why the CLSR and the RLSR need not yield the same ratio when the spectrum is completely real in the following manner: Without loss of generality let us assume that corresponding to the energy $E_i$ we have $E_i - E_{i-1} < E_{i+1} - E_i$ and we define $\Delta i, n = |E_i - E_{i+n}|$. The RLSR is then defined for $E_i$ as $\Delta_{i,-1}/\Delta_{i,1}$. However, it is possible that $E_i - E_{i-2} < E_{i+1} - E_i$ in which case the CLSR as defined for $E_i$ is $\Delta_{i,-1}/\Delta_{i,-2}$ since it involves the ratio of energy differences between an eigenvalue and its nearest and next nearest neighbors. We can thus clearly see that when averaged over all the energy eigenvalues, the CLSR is guaranteed to be greater than or at least equal to the RLSR. Furthermore, the values of the CLSR and RLSR that are obtained for the PTKR model in the chaotic limit of the Hermitian case match with those obtained from averaging over purely random matrices in the Gaussian Orthogonal Ensemble (GOE). The transition in the Hermitian case has been highlighted in the color plot as $T^*$ for aid in comparison.

We now consider the non-Hermitian case ($\lambda \neq 0$), which is of particular interest in this study. In Fig. 2, we show the behavior of the CLSR for varying values of kicking strength $K$ and the non-Hermiticity $\lambda$, computed for $\hbar_{\text{eff}} = 0.2$, and system-size $N = 6005$. We define $\alpha = \max\{\text{Im}(E)\}$ which only becomes non-zero when the spectrum starts to possess imaginary eigenvalues of the energy or, equivalently, when $\mathcal{PT}$-symmetry breaks (see supplementary material). We find that the phase diagram derived from the CLSR consists of three regimes: a $\mathcal{PT}$-symmetric integrable phase, a $\mathcal{PT}$-symmetric chaotic phase, and $\mathcal{PT}$-symmetry broken chaotic phase.

There are three possible transitions that can occur: 1) From the $\mathcal{PT}$-symmetric integrable phase to the $\mathcal{PT}$-symmetry broken chaotic phase, which we label $T_1$, 2) from the $\mathcal{PT}$-symmetric chaotic phase to the $\mathcal{PT}$-symmetry broken chaotic phase, which we label $T_2$ and 3) from the $\mathcal{PT}$-symmetric integrable phase to $\mathcal{PT}$-symmetric chaotic phase, which we label $T_3$. These are shown in Fig. 2. $T_3$ is similar to the transition seen in the Hermitian case. This is because, along $T_3$, we see that the CLSR goes from 0.50 to 0.57, which is exactly what is observed when going from the integrable to the chaotic regime for $\lambda = 0$. For transition $T_2$, we see a corresponding abrupt change in the value of $\alpha$ across the transition from the $\mathcal{PT}$-symmetric integrable regime to the $\mathcal{PT}$-symmetry broken chaotic regime. It can be seen that $\mathcal{PT}$-symmetry breaking at the transition $T_1$ does not occur abruptly as can be expected for a system with finite Hilbert space dimension (See Fig. 2). Therefore, the exact position of the transition point cannot be determined with great precision; however, the CLSR does show a clear transition. We define a threshold value of $\alpha$ above which the $\mathcal{PT}$-symmetry is assumed to be broken. The threshold, keeping in mind numerical errors, is taken to be $\alpha \leq 10^{-10}$ for unbroken $\mathcal{PT}$-symmetry. We observe that the CLSR takes on the same value of 0.57 in the $\mathcal{PT}$-symmetric chaotic regime as it does in the chaotic regime of the Hermitian model. However, we find that in the $\mathcal{PT}$-symmetric chaotic regime, the value of $\alpha$ can be greater than the threshold of $10^{-10}$, demonstrating that the highest imaginary value of the eigenenergies is not a particularly good diagnostic to detect the absence of $\mathcal{PT}$-symmetry.

It is interesting to note that the transition from the $\mathcal{PT}$-symmetric integrable phase to the $\mathcal{PT}$-symmetric chaotic also occurs across $T_3$. The hermitian case $\lambda = 0$ exhibits a transition

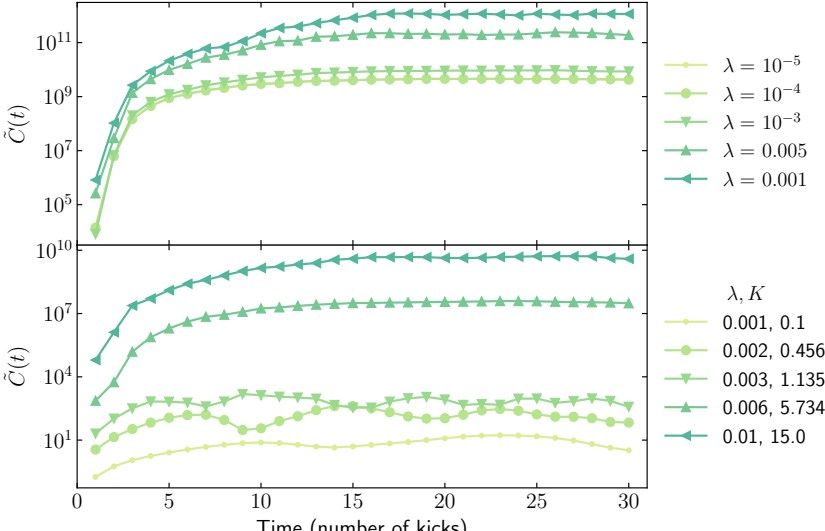

Figure 3: The normalized OTOC $\tilde{C}(t)$ vs $t$ across the transition along **Top** $T_1$ and **Bottom** $T_2$ shown in Fig. 2, calculated for system size $N = 2^{14}$. For $T_2$ (bottom), the values $(\lambda, K) = \{(0.001, 0.1), (0.002, 0.456), (0.003, 1.135)\}$ correspond to the $\mathcal{PT}$-symmetry unbroken integrable phase, while the values $(\lambda, K) = \{(0.006, 5.734), (0.01, 15.0)\}$ correspond to the $\mathcal{PT}$-symmetry broken chaotic phase. For $T_1$ (top), the values $(\lambda, K) = \{(10^{-5}, 30), (10^{-4}, 30), (10^{-3}, 30)\}$ correspond to the $\mathcal{PT}$-symmetry unbroken chaotic phase, while the values $(\lambda, K) = \{(0.005, 30), (0.001, 30.0)\}$ correspond to the $\mathcal{PT}$-symmetry broken chaotic phase.

from $\mathcal{PT}$- symmetric integrable to the $\mathcal{PT}$-symmetric chaotic phase as indicated by $T^*$. A similar transition can thus be expected when one varies $K$ for $\lambda$ sufficiently small but not equal to zero as seen in the left panel of Fig. 2. The same transition can also be effected by varying $\lambda$ but not $K$ as indicated by $T_3$. This is somewhat surprising as it seems to lack any Hermitian counterpart. One is thus able to induce chaos, as seen from the measured CLSR values, by increasing $\lambda$ whilst holding $K$ to the traditional integrable value. The nature of this transition is intriguing and requires further study, which we defer to future work.

## 4.2 OTOC

We present our results for the OTOC, focusing on its behavior across the different phases identified via the CLSR analysis. In the $\mathcal{PT}$-symmetry broken phase, the emergence of complex eigenvalues leads to unbounded growth of the state norm under time evolution. To account for this, we define a normalized OTOC (see Appendix D), which removes the exponential growth arising purely from non-Hermiticity and isolates the contribution due to chaotic dynamics.

$\mathcal{PT}$-symmetric integrable phase: The Hermitian case $\lambda = 0$ has been studied previously [42], where it was shown that the OTOC follows a power law for all time scales in the integrable phase. On the other hand, the OTOC features a transition from exponential growth at early times $t < t_E$ to a power law in the chaotic phase. In Fig. 3 (bottom), we show the normalized OTOC $\tilde{C}(t)$ for the $\mathcal{PT}$-symmetric integrable phase with parameters $(\lambda, K) \in \{(0.001, 0.1), (0.002, 0.456), (0.03, 1.135)\}$, and system-size $2^{14}$. We find that the normalized OTOC exhibits a behavior similar to the Hermitian case, which agrees with the calculations of the CLSR.

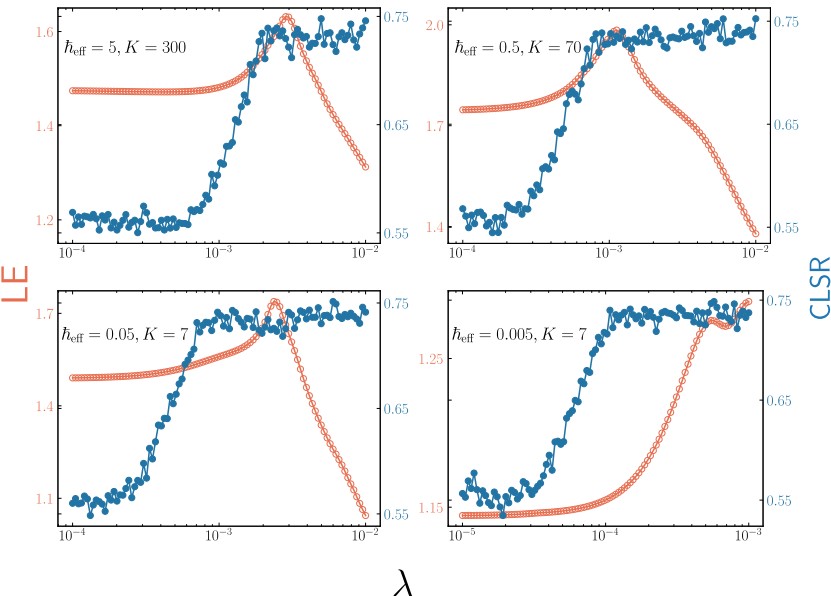

Figure 4: The above figure provides a comparison between the observed $\mathcal{PT}$-symmetric chaotic $\rightarrow$ PT-symmetry broken chaotic regime transition probed by the CLSR and the accompanying LE values for the same set of parameters for $K$ and $\lambda$. These have been plotted for various values of $\hbar_{\mathrm{eff}}$ that span three orders of magnitude. We observe that this transition observed by the CLSR is accompanied by a peak in the LE. Note that different values of $K$ have only been used to ensure the transition is properly captured, since, as previously stated, the $\mathcal{PT}$-symmetric chaotic regime shrinks as $\hbar_{\mathrm{eff}} \rightarrow 0$.

$\mathcal{PT}$-*symmetric chaotic phase*: Figure 3 (top) shows the normalized OTOC for the $\mathcal{PT}$-symmetric chaotic phase for $\lambda = (10^{-5}, 10^{-4})$, and fixed $K = 30$. As for the $\mathcal{PT}$-symmetric integrable phase, we find that the normalized OTOC in the symmetric chaotic phase exhibits behavior similar to that of the Hermitian system in agreement with CLSR calculations.

$\mathcal{PT}$-*symmetric broken chaotic phase*: This is the only $\mathcal{PT}$-symmetry broken phase present in the phase diagram. In Fig. 3 (top), we show the normalized OTOC for $\lambda = (10^{-3}, 0.005, 0.001)$, and fixed kicking strength $K = 30$. In Fig. 3 (bottom), the normalized OTOC for $(\lambda, K) \in \{(0.006, 5.734), (0.01, 15)\}$ is shown. Since the normalized OTOC eliminates the growth due to non-Hermiticity, it exhibits behavior similar to that in the $\mathcal{PT}$-symmetric unbroken chaotic phase. Below we the study behavior of the growth rate by calculating the Lyapunov exponent across the $T_2$ transition.

The normalized OTOC captures only the growth due to chaotic dynamics and thus allows for the extraction of a Lyapunov exponent (LE). Numerically, in order to extract the exponent from $\tilde{C}(t)$, we determine the times, after which the exponential growth starts slowing down, and fit $\tilde{C}(t)$ from $t = 1$ up to these times to the function $ae^{2\Lambda(t-1)}$ to find the parameter $\Lambda$. Numerical overflows prevent a calculation of the LE if it has a large value. We, thus, have to confine ourselves to a calculation of the exponent only in a region of the phase where $\mathcal{PT}$-symmetry is broken weakly.

In Figure 4, we show the LE as well as the corresponding CLSR across the transition $T_1$ for varying $\hbar_{\mathrm{eff}}$. We find that for all values of $\hbar_{\mathrm{eff}}$, the LE displays a peak. It is important to note that the peak does not appear at the transition $T_1$ but instead exists inside the broken

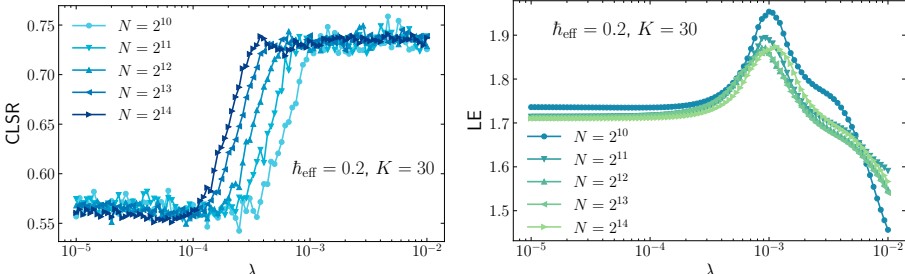

Figure 5: The figure above shows the transition from the PT-symmetric integrable regime to the PT-symmetric chaotic regime for different system sizes that increase by a factor of 2. $\hbar_{\text{eff}} = 0.2$ and $K = 30$ for all the plots. It can be seen that the transition, as obtained from the CLSR gets sharper as the system size ($N$) increases. There is a slight shift in the value of $\lambda$ at which the transition occurs and this shift appears to decrease with successively increasing values of $N$. The peak in the LE also appears to remain at a nearly fixed value of $\lambda$ as the system size is varied.

phase of the $\mathcal{PT}$-symmetry. A further investigation of the nature of the peak in the Lyapunov exponent profile (See Sec. 5) is required. We also note that in a calculation performed on the same system as ours [59], a different definition of the OTOC has been used, one involving both the operators $\theta$ and $p$. The OTOC defined this way displays polynomial growth in time in the $\mathcal{PT}$-symmetry broken regime and not an exponential one, as we observe.

## 4.3 Parameter dependence

We summarize here the behavior of the CLSR and OTOC as we take three particular limits of interest.

- $N \to \infty$: This is the limit we take to cover the ring of allowed eigenvalues of $\hat{p}$ densely and approach the continuum limit. The CLSR and LE plots show little to no noticeable shift when increasing the value of $N$, as can be seen from Fig. 5. The phase transitions seen on the CLSR become sharper but still occur roughly at the same value of $\lambda$ with a slight drift.

- $\lambda \to 0$: This limit signifies the Hermitian limit of the problem where only two regimes have been identified with the CLSR values 0.5 and 0.57 corresponding to the integrable and chaotic regimes, respectively. These correspond to the $\mathcal{PT}$-symmetric integrable regime and $\mathcal{PT}$-symmetric chaotic regime for the general ($\lambda \neq 0$) case. It can be seen from Fig. 6 that the value of $K$ for the transition between the two phases in the Hermitian limit persists even for small values of $\lambda$ for finite values of $\hbar_{\text{eff}}$.

- $\hbar_{\text{eff}} \to 0$: This limit maps the quantum problem to its classical counterpart where again, there exists a phase transition from the integrable phase to a chaotic phase. As can be seen in Fig. 6, as $\hbar_{\text{eff}} \to 0$, the transition from $\mathcal{PT}$-symmetric intergrable to $\mathcal{PT}$-symmetric chaotic is happening at $K \approx 1$, which is where the classical transition occurs [60]. This observation is also in agreement with the transition observed for the Hermitian quantum kicked rotor [61] as $\hbar_{\text{eff}} \to 0$. Further, we observe the shrinking of the $\mathcal{PT}$-symmetric chaotic regime as $\hbar_{\text{eff}} \to 0$. From the Hermitian case, we know that the $\mathcal{PT}$-symmetric chaotic regime will always exist on the $\lambda = 0$ line. However, we postulate that in the limit as $\hbar_{\text{eff}} \to 0$, the only phase present in the $\lambda \neq 0$ space will be the $\mathcal{PT}$-symmtric chaotic unbroken phase, i.e, an infinitesimal value of $\lambda$ will be sufficient to break $\mathcal{PT}$-symmetry, leading to chaotic nature in this limit.

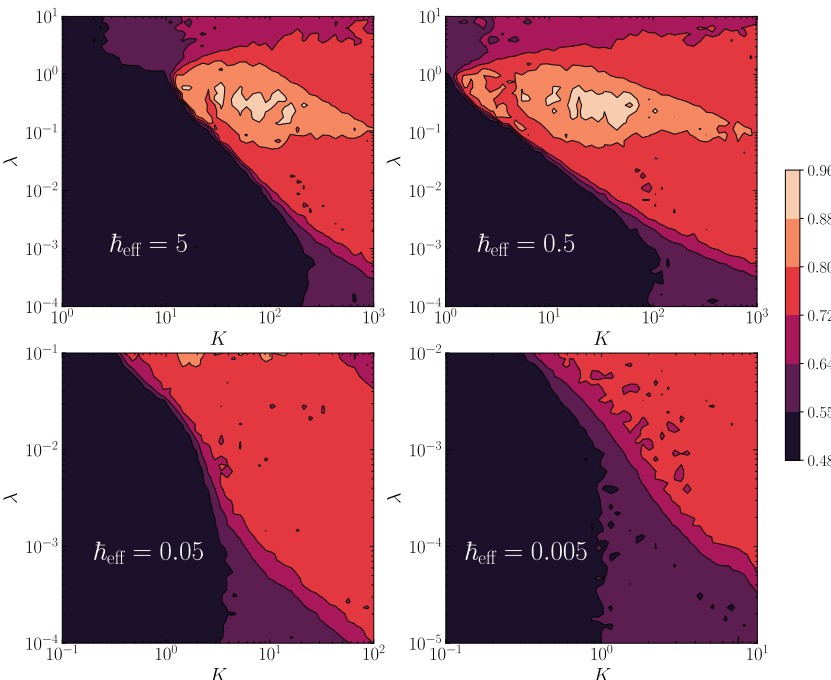

Figure 6: The above figure shows a section of the CLSR contour map for different values of $\hbar_{\text{eff}}$. It can be observed that the transition $T^*$ moves closer to the integrable $\rightarrow$ chaotic transition seen in the classical model for which $K \sim 1$ as $\hbar_{\text{eff}} \rightarrow 0$. Furthermore, the $\mathcal{PT}$-symmetric chaotic regime is observed to shrink in size with a decrease in $\hbar_{\text{eff}}$

## 5   Conclusions

We have performed a comprehensive numerical study of the non-Hermitian kicked rotor demonstrating, for the first time (to our knowledge), the presence of integrability and $\mathcal{PT}$-symmetry breaking transitions in a driven quantum model. The model we have studied exhibits three phases, i) A $\mathcal{PT}$-symmetric integrable phase, ii) A $\mathcal{PT}$-symmetric chaotic phase and iii) A $\mathcal{PT}$-symmetry broken chaotic phase, thus demonstrating that $\mathcal{PT}$-symmetry breaking is a sufficient condition for the onset of chaos. We have characterized these phases and the transitions between them by calculating the complex level spacing ratio (CLSR) and the out of time ordered correlator (OTOC).

The CLSR phase diagram (see Fig. 2) elucidates several properties of the PTKR model and the intricate relation of $\mathcal{PT}$-symmetry breaking and chaos. Foremost, it shows that the CLSR is a viable diagnostic to differentiate the several phases in our system for both the non-Hermitian and Hermitian cases. One of the main observations is the absence of an integrable, $\mathcal{PT}$-symmetric phase in the phase diagram as determined from the complex level spacing ratio $\langle r \rangle$, which implies the sufficiency of $\mathcal{PT}$-symmetry breaking for the setting in of chaos. Equivalently, this indicates the absence of a $\mathcal{PT}$-symmetry broken, integrable phase in our model. Note however that the model is integrable with broken $\mathcal{PT}$ symmetry when $\lambda = \infty$ and $m = \infty$. In this limit, the Floquet eigenvalues are simple $iK \sin \theta$ for all $\theta \in [0, 2\pi)$. Thus, the eigenvalues are pure imaginary, occur in complex conjugate pairs showing that the system breaks $\mathcal{PT}$ symmetry. Moreover, since they can also be obtained exactly, the system is clearly integrable. However, it appears from our numerical study that this special $\mathcal{PT}$ symmetry broken integrable point does not extend into a phase for finite $m$ and $\lambda$. The question of whether the absence of such a phase is special to our model or more generic will require further

investigation. We have also made a few other other interesting observations, which, while not very clearly understood at this stage, motivated further work. The first is that a transition from between the $\mathcal{PT}$-symmetric integrable and chaotic phase can be effected by varying only the non-Hermiciticy parameter $\lambda$. This is quite intriguing since, naively, one might expect that such a transition would necessarily require varying the kicking strength, such as in the purely Hermitian system. The second observation is the appearance of a peak in the Lyapunov exponent as a function of the non-Hermiticity parameter $\lambda$ inside the $\mathcal{PT}$-symmetry broken phase close to the transition. A broad peak in the Lyapunov exponent has been observed in a Hermitian Bose-Hubbard in the vicinity of a quantum phase transition [32] but further investigation is required to determine whether or not there is any connection between the aforementioned observation and ours.

We obtain the Lyapunov exponent from a calculation of the OTOC. We show that the early time growth of the OTOC can be used to distinguish between the three phases we observe. In particular, once we define a normalized version of the OTOC to eliminate the effects of the growth in time due to the complex nature of eigenvalues, it can be employed to detect the transition from the $\mathcal{PT}$-symmetric chaotic phase to the $\mathcal{PT}$-symmetry broken chaotic phase. We also observe that the reality of the eigenvalues in the above two regimes does not display a sharp disappearance at the transition, so the origin of the sharp jump in the CLSR, which indicates the transition between the two phases, needs further investigation.

# Acknowledgements

H.S. would like to thank Aranya Bhattacharya for useful discussions. SM thanks the DST, Govt. of India for support.

# A    Numerics for the level spacing calculations

This section describes the computational schemes employed along with the necessary theoretical motivation.

## A.1    Modifying the Hamiltonian

The Hamiltonian of a $\mathcal{PT}$-symmetric kicked rotor is conventionally chosen to have the following form for the potential $V(\theta)$.

$$V(\theta) = K(\cos\theta + i\lambda\sin\theta) \tag{10}$$

However, in order to be able to simultaneously study the effect of varying the kicking strength $K$ and the non-Hermiticity parameter $\lambda$, the above form is not a good choice. For $\lambda \gg 1$, we see that $\lambda$ not only tunes the non-Hermiticty parameter but also controls the overall 'magnitude' of the $V(\theta)$ term and so controls the kicking strength as well. To avoid this, we 'normalize' the term inside and use the following definition for $V(\theta)$.

$$V(\theta) = \frac{K}{\sqrt{1+\lambda^2}}(\cos\theta + i\lambda\sin\theta) \tag{11}$$

## A.2    Accounting for degeneracies due to symmetries

While calculating the energy level spacing distribution from the eigenvalue spectrum of a Hamiltonian, it is very important that one accounts for degeneracies arising from symmetries

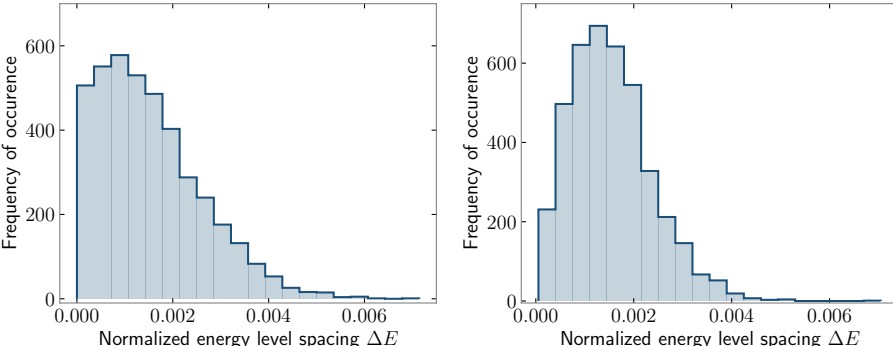

Figure 7: The level spacing distribution for $k \simeq 100$. It can be seen that due to degeneracies arising from symmetries or near-symmetries, the distribution does not match the GOE form (left panel). The distribution matches the GOE form after introducing a random small perturbation of $10^{-3}m$ (right panel).

of the Hamiltonian. Hence, it is a common practice to carry out the level spacing calculations on a subset of the set of eigenvalues without repeating degenerate eigenvalues. The Hermitian kicked rotor Hamiltonian has parity $\mathcal{P}$ as one of its symmetries. To achieve such a subset, we block diagonalize the Hamiltonian using a basis that diagonalizes the group of symmetries of the Hamiltonian. Then, we perform the level spacing calculations over one of the boxes of the Hamiltonian.

In our case, defining the Floquet operator and retrieving the eigenvalues results in a large number of unidentified symmetries and near symmetries, causing a peak to exist near $\Delta E$ (level spacing) $= 0$ in the distribution, as can be seen in figure 7. To improve the statistics and get rid of such near symmetries, we modify the kinetic energy term in the Hamiltonian. When written in the angular momentum basis, the kinetic energy is diagonal with its diagonal elements being $-\dfrac{\hbar^2 l^2}{2m}$ where $l$ is the angular momentum eigenvalue and $m$ the moment of inertia. To break such symmetries, we add randomness to the masses of each diagonal term individually by redefining the diagonal elements as $-\dfrac{\hbar^2 l^2}{2(m + \delta m)}$. For our calculations, $m = 1$ and the $\delta m$'s are chosen randomly from a uniform distribution over the interval $[0, 10^{-3}]$ for each diagonal element. This has the desired effect while still preserving the transition from the integrable to the chaotic phase. Figure 7 provides an example of this in the Hermitian case. The Hermitian version of the PTKR model has $\mathcal{P}$ and $\mathcal{PT}$ as symmetries of the system. The plot is of the Wigner-Dyson level spacing distribution.

## A.3 Unfolding procedure

Whether using real or complex eigenvalues, the unfolding process is crucial to getting a faithful representation of the energy level spacing distribution. The energy level spacing has the dimensions of energy. Wigner's analysis of the level spacing distributions involved calculations over a large ensemble of $2 \times 2$ matrices with particular symmetries. Suppose that this is also true for the distribution obtained from an $N \times N$ random matrix with the same symmetries over its spectrum. In this case, it must be that the level spacing in different parts of the spectrum is not related or weakly related. Consequently, in order to compare level spacing across distinct sections of the spectrum, it becomes necessary to work with dimensionless quantities. These quantities can be obtained by dividing the level spacing by the mean level spacing in a small vicinity surrounding them. Such a procedure is known as unfolding.

For a real spectrum, the mean is simply obtained by averaging the level spacing about that

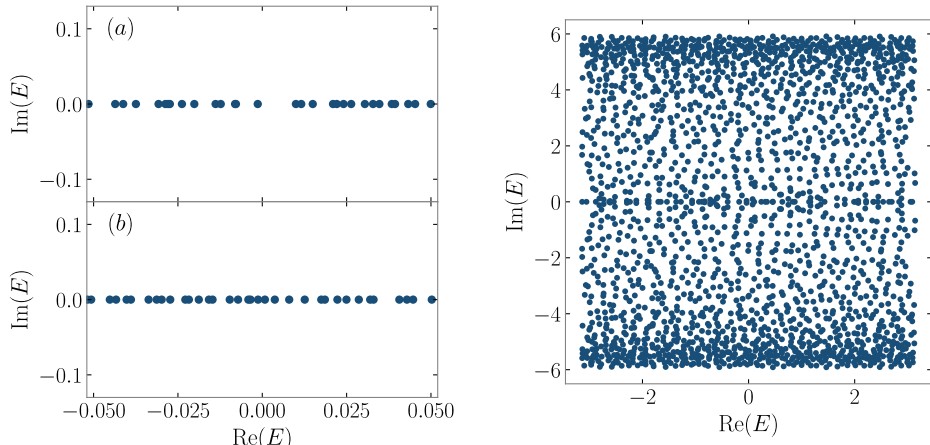

Figure 8: The above figure represents depicts the spread of the quasienergies in the three different regimes. Top left : $\mathcal{PT}$ symmetric integrable, Bottom left : $\mathcal{PT}$ symmetric chaotic, Right : $\mathcal{PT}$ symmetry broken chaotic.

eigenvalue for a fixed radius of points. For a complex spectrum, we define $\frac{1}{\sqrt{\rho_k}}$ as the mean level spacing around an eigenvalues $\xi$. Here $\rho_k$ is defined as follows.

$$\rho_k = \frac{3n}{\pi \left( r_{k,n-1}^2 + r_{k,n}^2 + r_{k,n+1}^2 \right)} \tag{12}$$

Above, $n$ is predefined depending on our Hilbert space dimensions $N$. In our calculations, $n = 10$ appears to work well for $N = 2001$. Note that real or complex level spacing ratios do not need to undergo unfolding procedures since they already involve averaging over dimensionless quantities.

## B    Calculation on random matrix ensembles and new CLSR

In our work, we show the agreement of the level spacing ratio obtained from our model and that from calculations over random matrices with the same symmetries as those of the model. Some of these values have been calculated previously as well [55]. We perform random matrix calculations for complex universality classes GinUE, GinOE, and AI$^\dagger$ [53]. Here, we highlight the method used to perform such calculations.

First, a complex matrix with $2N^2$ real entries is defined, giving complex random matrices of size $N \times N$. Every real entry is independently and identically chosen from a normal distribution of mean 0 and standard deviation 1. Any further symmetry constraints are added as follows: Let the symmetry group of these constraints be $\mathcal{A}$, then all the elements of $\mathcal{A}$ are made to act on copies of the same random matrix, and all of the resultant matrices are added. Next, the eigenvalues of the matrix thus obtained are numerically calculated, and the CLSR is found. This process is repeated enough times so as to ensure that the standard deviation of all the measures is sufficiently small compared to the mean.

The complex level spacing ratio can also be defined using angles, which is the other definition provided in [54]. This definition has not been adopted by us since it fails for a real eigenspectrum where $\theta$ becomes ill-defined. However, we perform random matrix calculations for $-\langle \cos \theta \rangle$ as well as described below.

Table 2: The table below consists of data gathered on the complex level spacing ratio for some complex universality classes (GinUE, GinOE, and AI$^\dagger$); bracket contains must be read as (mean value, standard deviation).

| Universality class | $\langle r \rangle$ | $-\langle \cos\theta \rangle$ |
|---|---|---|
| A/GinUE | (0.73809, 0.00207) | (0.23297, 0.0093) |
| GinOE | (0.73809,0.0048) | (0.2364,0.0131) |
| AI$^\dagger$ | (0.7231, 0.0027) | (0.18777, 0.0077) |

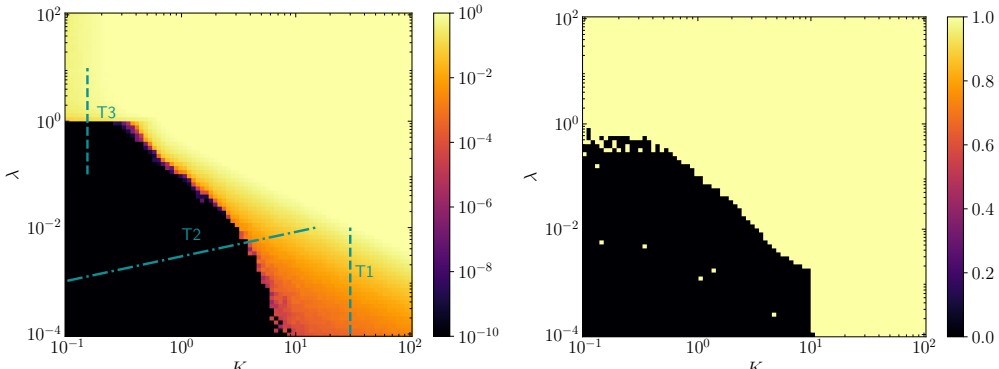

Figure 9: **Left**: The maximum of imaginary part $\alpha$ of the eigenspectrum for varying values of the kicking strength $K$ and the non-hermiticity $\lambda$, computed for $N = 4096$. The dotted lines $T_1$, $T_2$, and $T_3$ represent the transition across the $\mathcal{PT}$-symmetric chaotic phase to the $\mathcal{PT}$ symmetry broken chaotic phase, $\mathcal{PT}$-symmetric integrable phase to the $\mathcal{PT}$-symmetry broken chaotic phase, and the $\mathcal{PT}$-symmetric integrable phase to the $\mathcal{PT}$-symmetry broken chaotic phase. **Right**: The color plot highlights the region of the phase diagram for which the CLSR is close to 0.5 which corresponds to the $\mathcal{PT}$ symmetric integrable phase.

## C $\quad \mathcal{PT}$-symmetry broken phases

The quasienergy of the PTKR model is complex i.e. $\epsilon = \epsilon_r + i\epsilon_i$. When the maximum value of the imaginary part (denoted by $\alpha$) exceeds a certain threshold value $\alpha_c$, we assume the $\mathcal{PT}$-symmetry to be broken. The yellow shaded region in fig.9 is the $\mathcal{PT}$ symmetry broken region. In parts of the region near the transitions, we obtain a CLSR close to 0.739. This is the value of the CLSR one obtains for the GinOE universality class, to which the Hamiltonian belongs. This is due to the fact that the Hamiltonian is not bound by Hermiticity, and it commutes with an anti-unitary operator (namely $\mathcal{PT}$). In the table 3, we show calculations of the CLSR on random matrices in the GinOE class. Additionally, we also perform calculations on matrices within GinOE that commute with $\mathcal{PT}$. These are in agreement with the value of the CLSR we get in our phase diagram.

## D $\quad$ Unnormalized OTOC and Norm Growth

In the $\mathcal{PT}$-symmetry broken phase, the eigenspectrum consists of complex eigenvalues. As a result, the time-evolved state $|\Psi(t)\rangle = U(t)|\Psi\rangle$ no longer remains normalized (see Fig. 10). The late-time growth of both the OTOC and the state norm depends on the eigenspectrum —

Table 3: The table below consists of data gathered on the CLSR for the $\mathcal{PT}$ symmetry broken chaotic regime and its comparison with CLSR in the complex universality classes GinOE and its subset of matrices that commute with $\mathcal{PT}$, bracket contents must be read as (mean value, standard deviation).

| Universality class | CLSR |
|---|---|
| Phase diagram | (0.735) |
| GinOE | (0.73809, 0.0048) |
| $\mathcal{PT}$-symmetric matrix | (0.73942, 0.0054) |

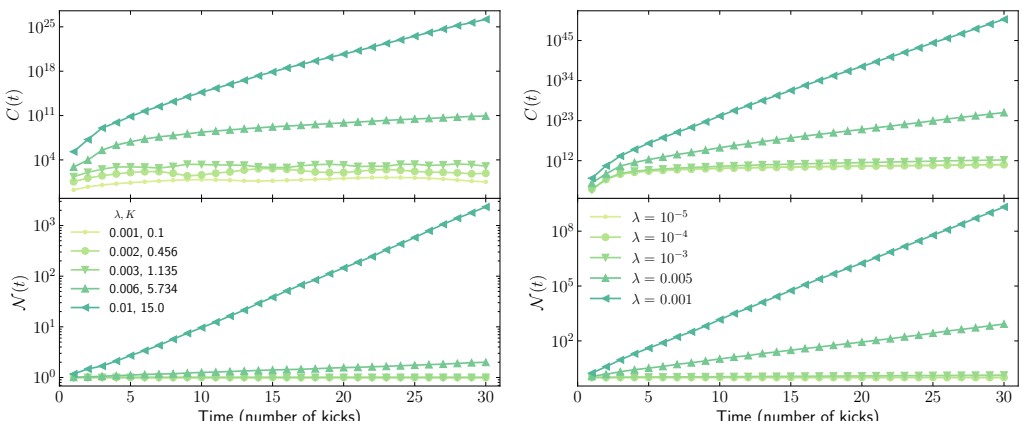

Figure 10: The OTOC $C(t)$ and state norm $\mathcal{N}(t)$ vs $t$ across the transition along **Left** $T_2$ and **Right** $T_1$ shown in Fig. 2, calculated for system size $N = 2^{14}$. For $T_1$ (Left), the values $(\lambda, K) = \{(0.001, 0.1), (0.002, 0.456), (0.003, 1.135)\}$ correspond to the $\mathcal{PT}$-symmetry unbroken integrable phase, while the values $(\lambda, K) = \{(0.006, 5.734), (0.01, 15.0)\}$ correspond to the $\mathcal{PT}$-symmetry broken chaotic phase. For $T_2$ (right), the values $(\lambda, K) = \{(10^{-5}, 30), (10^{-4}, 30), (10^{-3}, 30)\}$ correspond to the $\mathcal{PT}$-symmetry unbroken chaotic phase, while the values $(\lambda, K) = \{(0.005, 30), (0.001, 30.0)\}$ correspond to the $\mathcal{PT}$-symmetry broken chaotic phase. In both cases, the $\mathcal{PT}$-symmetric broken phase displays a late-time exponential growth of the OTOC and the state norm.

in particular, it depends strongly on the maximum of the imaginary part of the eigenvalues of the Hamiltonian. Let us denote this maximum value by $\alpha$. At late times, the evolution is dominated by the eigenvalue with the largest imaginary part:

$$U(t \to \infty)|\Psi\rangle \propto e^{\alpha t}|\Psi\rangle, \tag{13}$$

Consequently, the late-time norm $\mathcal{N}(t) = \langle\Psi(t)|\Psi(t)\rangle$ scales as:

$$\mathcal{N}(t \to \infty) \propto e^{2\alpha t}. \tag{14}$$

To analyze the behavior of the OTOC, we expand the squared commutator $-[\hat{p}(t), \hat{p}(0)]^2$ as

$$\hat{p}(0)\hat{p}(t)\hat{p}(t)\hat{p}(0) + \hat{p}(t)\hat{p}(0)\hat{p}(0)\hat{p}(t) - \hat{p}(t)\hat{p}(0)\hat{p}(t)\hat{p}(0) - \hat{p}(0)\hat{p}(t)\hat{p}(0)\hat{p}(t) \tag{15}$$

In the large-$t$ limit, the contribution of each term scales as:

$$\langle\Psi|\hat{p}(t)\hat{p}(0)\hat{p}(t)\hat{p}(0)|\Psi\rangle \propto e^{4\alpha t} \tag{16}$$

Hence, the unnormalized OTOC exhibits exponential growth:

$$C(t \to \infty) \propto e^{4\alpha t} \tag{17}$$

This motivates the definition of a normalized OTOC:

$$\tilde{C}(t) = e^{-4\alpha t} C(t). \tag{18}$$

which removes the exponential divergence associated with the non-unitary dynamics. The normalized OTOC $\tilde{C}(t)$ thus captures only the growth associated with the chaotic nature of the system.

Since our calculations are performed at finite times, subleading eigenvalues can still contribute to the dynamics. Therefore, the normalization factor $\alpha$ is determined by fitting the late-time growth of $C(t)$ to an exponential form.

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
