# Peer review of "Quantum chaos in PT symmetric quantum systems"

_SciPost Physics, doi:SciPost Phys. 19, 120 (2025)_

## Round 1 · Referee Report · Anonymous (Referee 1) · 2025-9-29

Report

The authors have responded to my questions and have made extensive
corrections to the paper as requested. I think the paper can now be published as it.

Recommendation

Publish (meets expectations and criteria for this Journal)

---

## Round 1 · Referee Report · Anonymous (Referee 2) · 2025-10-2

Report

The authors have adequately responded to all my comments and criticisms, and made necessary changes in the revised manuscript appropriately. The current version of the manuscript can be published in SciPost Physics.

I would request the authors of this article to make their numerical code and data publicly available via GitHub or Zenodo or any other platform that they find appropriate. Accordingly, the authors should add a statement on "Data availability" in the manuscript.

Recommendation

Publish (meets expectations and criteria for this Journal)

---

## Round 1 · Author Response

Dear Editor,
Response to the referees
Thank you very much for sending our paper out for review and we apologize for the delay in resubmission.
We are happy to note that referee 2 finds no weaknesses in our paper and says that the numerical results are “sufficiently clear and convincing”, the results are presented “in a coherent fashion” and that “the authors also point out the open questions based on their findings”, and recommends publication in SciPost Phys. They suggest the addition of certain references, which we have done in the revised manuscript.
We are also happy to note that referee 1 finds one of the main results of our work that PT symmetry breaking results in chaos, interesting. They say that “despite the interesting result, the paper needs a bit of a makeover and the following questions/points need to be addressed before any recommendation for publication can be made”. In this resubmission, we have addressed all the questions and comments of the referee in detail. We believe that the comments of the referee have helped us improve the presentation of our work in this resubmission and than them for the same.
We have already uploaded a PDF with a point by response to the comments of both referees to the page with the original submission. We hope that this will suffice and look forward to the publication of our work in SciPost.
Thank you very much.
With best regards,
Kshitij Sharma, Himanshu Sahu and Subroto Mukerjee

---

## Round 1 · List of Changes

Changes pertaining to the referees' comments

REFEREE #1
Referee Comment 1
Please briefly elaborate on Wigner’s surmise that is mentioned in the first paragraph.

Changes: We have added a small discussion on Wigner’s surmise in the introduction (lines 66-69).

Referee Comment 2
Page 2: the authors motivate the paper as an extension of previous works. It would be useful to describe what the previous results are and highlight the contribution of the current work.

Changes: A motivation has been added in introduction — lines 119-122 and 124-128.

Referee Comment 3
Page3: Define zK , is K the strength ? Important as this forms the backbone of the work. How does one interpret this definition in a manner akin to normal level statistics for Hermitian systems?

Changes: In revised version, we changed the notation to represent zγ to avoid confusion.

Referee Comment 4
More info on the values for CLSR for Poisson and GOE. . . one does not get any sense of why these values are higher for PT symmetric matrices compared to the case of real symmetrical matrices.

Changes: A discussion has been added in Sec. 4.1 from lines 210 to 217.

Referee Comment 5
In Fig1, left caption: mention lambda=0, in the right figure, though RLSR is mentioned in the caption. . . there is no RLSR plotted

Changes: We thank the referee for pointing this out and have now made the suggested corrections.

Referee Comment 6
Pg.3 There are two computational techniques to improve the statistics in our calculation. The first is to replace m in Eq. (1) by m + ∆mp, where ∆mp is a small random number selected independently for each p.
Please indicate what this p corresponds to. It is discussed in the appendix and it would make sense to have a part of those statements repeated in the main body of the paper. I do not see the impact of App A.1 where the coupling strength is renormalized . . . as the authors really do not study the regime λ/K ≫ 1 in this work. So is this necessary ?

Changes: As suggested by the referee we have clarified what p is in the main text in section 3.1 (lines 180-183).

Referee Comment 7
Page 5. However, in this limit, Eq. 7 does not yield the standard real level spacing ratio (RLSR) for a real spectrum. Given the plots in Fig.1, where the CLSR approaches the RSLR values for GOE etc for the Hermitian case, could the authors try and explain why the CLSR as given by Eq. 7 give different values when there is nonhermiticity ? If zK only involves the Floquet spectrum which is real in the PT symmetric case, I do not understand what leads to these differences. In absence of any discussion on how zK is defined, it is very opaque. A brief explanation of why one averages over the kicking strength will be useful. I think the authors should elaborate on these points. A few plots of the quasienergy spectrum in the appendices would be illuminating.

Changes: We have already addressed this point in our Changes to ‘Commment 3’ of the referee. As suggested by them, we have now added a few plots of the quasi-energy spectrum to the appendix (Fig. 08) for points in different parts of the phase diagram.

Referee Comment 8
Page 6. Regarding the different lines in Fig2, T1, T2 etc. . . is the boundary where PT sym- metry breaking is observed a numerically obtained one by considering the Floquet eigenvalues or is there some analytical reasoning which can explain this ? Have the authors looked at possible similarity transformations of the Hamiltonian which gives rise to a pseudo-hermitian Hamiltonian, as this would help indicate the boundary between PT symmetric and PT broken regimes. This is something that one often does in non-hermitian problems (cf. Bender books and papers that the authors have referenced) Such a calculation will eminently help improve the clarity of the paper. Or is there some simplifcation that can be done in the limit ħ → 0 ? I would like the authors to address this.

Changes: No changes required.

Referee Comment 9
Page 6: Caption: Right: The absolute value of the maximum imaginary part of an energy eigenvalue, α, across the transition Main panel: T1 Inset T2, calculated for N = 4095. It can be seen that while α shows an abrupt change along T2, it seems to increase smoothly along T1. The CLSR on the other hand, shows an abrupt transition along both T1 and T2. This is very confusing as, there is a rather sharp change (within numerical finite size effects) in alpha along the T1 line as PT symmetry is broken. One expects α to show a non-analytic behaviour as the system crosses an exceptional point. In the inset, along T2, there is only one point which is off the smooth straight line behaviour of alpha. The authors seem to be drawing the opposite conclusion. I would appreciate a clarification of this.
Furthermore, in the inset for alpha what is varied across the x-axis ? The caption mentions that it is the variation along the T2 line, but there both lambda and K vary...so this is confusing. Additionally, the scales in the inset do not conform to the color map axes.

Changes: We thank the referee for pointing this out and apologize for the lack of clarity. In the revised version, the Fig. 02 captions are revised and now are consistent.

Referee Comment 10
Table1 caption . . . please mention that this is valid for the PT symmetric case only. In fact, I would recommend that the authors discuss the complex random matrix ensembles discussed in the appendix here and link it to the numerical results that obtain for the CLSR. Such a discussion would be very useful to contextualize the numerical results.

Changes: Table 03 is being added.

Referee Comment 11
In Fig. 1, we show the mean RLSR as well as the CLSR with varying values of the kicking strength K, for ħ = 0.2 and system-size N = 8005. We notice that the transition points are reasonably independent of the value of N. Thus, in what follows, the system size is chosen to be large enough so that all quantities are well converged. We find that both the CLSR and RLSR display a transition at the same value of K. In Fig.1, why does the deviation from Poisson statistics happens at quite different K values in the Hermitian case? Is there any understanding of this that the authors can provide ?

Changes: Figure 01 is updated with data shown for larger system size N = 10095.

Referee Comment 12
In Figs.3 and 4. I think the caption misstates which figures correspond to T1 and T2. This again confuses the reader.

Changes: We have changed the labels to the correct ones now.

Referee Comment 13
In all the parameter regimes shown in the normalized OTOC, the long time behaviour ap- proaches a constant. . . contrary to the authors; statement, this seems to be true irrespective of whether it is chaotic/PT broken or not. Could this authors discuss this more ?
Changes: The section 4.2 on OTOC has been revised to avoid this confusion and to move the unnormalized plots to appendix. In particular, we made sure to focus on early time behavior when comparing between different phases.

Referee Comment 14
In Fig.5, it would make for easier reading if all the axes had the same ranges to see the points made by the authors.
Changes: No changes required.

Referee Comment 15
The main result of the paper is that PT breaking is always accompanied by chaos. Can the authors address why there is no PT broken phase without chaos ? Could this plausibly be a feature of the particular model studied ? In the random matrix ensembles that the authors allude to for complex matrices and eigenvalue structures, is there a possibility to have integrability to chaos transitions or does RMT prohibit these ? As this is the principal result of the work, the authors should present some arguments, atleast heuristic to bolster their results.

Changes: We have added a discussion in the conclusion section discussing this point.

Referee Comment 16
Page 9: Given that non-unitary evolutions are known to not preserve state norms, I recom- mend only plotting the normalized OTOC in the main body of the paper and move Fig 3 with the Norm and OTOC to the supplemental material.

Changes: In revision, we have moved the plots and discussion on unnormalized OTOC and norm to appendix D. The main text (sec. 4.2) now only has discussion on normalized quantities.

Referee Comment 17
Page 13. . . . PT symmetric phase in the phase diagram as determined from the complex level spacing ratio 〈r〉, which implies the sufficiency of PT symmetry breaking for the setting in of chaos. We also made a couple of other interesting observations, which, while not very clearly understood at this stage, motivated further work.
I think the authors mean that an integrable PT broken phase is not present in this model.

Changes: We added lines 341-349 (in Sec. 05).

Referee Comment 18
Page 14: typo formafter
Changes: All typo has been corrected to best of our knowledge.

Referee Comment 19
Page 15. . . missing reference Some of these values have been calculated previously as well [?].

Changes: The missing citation has been added.

Referee Comment 20
In Table 2, GinOE is missing . . .

Changes: Table 02 has been revised to add data for GinOE.

Referee Comment 21
Table 3...Is this table indicating that PT broken and PT symmetric chaotic regimes are described by the same RMT ? If yes, this is very curious as in quantum problems with no chaos, PT broken regimes have very different behaviours of observables as compared to PT symmetric regimes. So if the onset of chaos blurs any difference, this would be a very important point to make. Are there other observables/quantities that one can numerical obtain for the PTKR that explore this further ? The authors should comment on this.

Changes: We would like to clarify that Table 3 only shows that if a random matrix is chosen that possesses PT symmetry (which our Hamiltonian does independent of the value of λ and K) then the resulting CLSR values are in agreement with those obtained from a larger set of random matrices in the GinOE category. PT- symmetric here only implies that the matrices have PT symmetry, not that their spectrum is such that they are in the PT symmetric phase.

Referee Comment 22
Appendix C PGE 16 The yellow shaded region is the PT symmetry broken region. . . . which figure are the authors referring to ? Cannot find yellow regions in the figures presented.

Changes: The missing plots (Fig. 09 in revised manuscript) has been added.

REFEREE #2

Referee Comment 1
In the sixth paragraph of the Introduction, the authors discuss applicability of non-Hermitian Hamiltonians in various other context, such as topological phases of matter. In this context, I request them to include effects of e-e interactions, quantum criticality, and Lorentz symmetry in NH systems, which have been studied in the recent past. Some of the early works in this field include Physical Review Letters 132, 116503 (2024), Communications Physics 7, 169 (2024), Journal of High Energy Phys. 01, 143 (2024), SciPost Phys. 18, 073 (2025)

Changes: We thank the referee for their input and have added the suggested references and discussion in the introduction.

---

## Editorial Decision

published